# Learning Koopman Invariant Subspaces for Dynamic Mode Decomposition

**Naoya Takeishi[§], Yoshinobu Kawahara[†,‡], Takehisa Yairi[§]**
[§]Department of Aeronautics and Astronautics, The University of Tokyo
[†]The Institute of Scientific and Industrial Research, Osaka University
[‡]RIKEN Center for Advanced Intelligence Project
{takeishi,yairi}@ailab.t.u-tokyo.ac.jp, ykawahara@sanken.osaka-u.ac.jp

## Abstract

Spectral decomposition of the Koopman operator is attracting attention as a tool for the analysis of nonlinear dynamical systems. Dynamic mode decomposition is a popular numerical algorithm for Koopman spectral analysis; however, we often need to prepare nonlinear observables manually according to the underlying dynamics, which is not always possible since we may not have any *a priori* knowledge about them. In this paper, we propose a fully data-driven method for Koopman spectral analysis based on the principle of *learning Koopman invariant subspaces* from observed data. To this end, we propose minimization of the residual sum of squares of linear least-squares regression to estimate a set of functions that transforms data into a form in which the linear regression fits well. We introduce an implementation with neural networks and evaluate performance empirically using nonlinear dynamical systems and applications.

## 1 Introduction

A variety of time-series data are generated from nonlinear dynamical systems, in which a state evolves according to a nonlinear map or differential equation. In summarization, regression, or classification of such time-series data, precise analysis of the underlying dynamical systems provides valuable information to generate appropriate features and to select an appropriate computation method. In applied mathematics and physics, the analysis of nonlinear dynamical systems has received significant interest because a wide range of complex phenomena, such as fluid flows and neural signals, can be described in terms of nonlinear dynamics. A classical but popular view of dynamical systems is based on state space models, wherein the behavior of the trajectories of a vector in state space is discussed (see, e.g., [1]). Time-series modeling based on a state space is also common in machine learning. However, when the dynamics are highly nonlinear, analysis based on state space models becomes challenging compared to the case of linear dynamics.

Recently, there is growing interest in operator-theoretic approaches for the analysis of dynamical systems. Operator-theoretic approaches are based on the Perron–Frobenius operator [2] or its adjoint, i.e., the Koopman operator (composition operator) [3], [4]. The Koopman operator defines the evolution of observation functions (observables) in a function space rather than state vectors in a state space. Based on the Koopman operator, the analysis of nonlinear dynamical systems can be lifted to a linear (but infinite-dimensional) regime. Consequently, we can consider modal decomposition, with which the global characteristics of nonlinear dynamics can be inspected [4], [5]. Such modal decomposition has been intensively used for scientific purposes to understand complex phenomena (e.g., [6]–[9]) and also for engineering tasks, such as signal processing and machine learning. In fact, modal decomposition based on the Koopman operator has been utilized in various engineering tasks, including robotic control [10], image processing [11], and nonlinear system identification [12].

One of the most popular algorithms for modal decomposition based on the Koopman operator is dynamic mode decomposition (DMD) [6], [7], [13]. An important premise of DMD is that the target dataset is generated from a set of observables that spans a function space invariant to the Koopman operator (referred to as Koopman invariant subspace). However, when only the original state vectors are available as the dataset, we must prepare appropriate observables *manually* according to the underlying nonlinear dynamics. Several methods have been proposed to utilize such observables, including the use of basis functions [14] and reproducing kernels [15]. Note that these methods work well only if appropriate basis functions or kernels are prepared; however, it is not always possible to prepare such functions if we have no *a priori* knowledge about the underlying dynamics.

In this paper, we propose a *fully data-driven* method for modal decomposition via the Koopman operator based on the principle of *learning Koopman invariant subspaces* (LKIS) from scratch using observed data. To this end, we estimate a set of parametric functions by minimizing the residual sum of squares (RSS) of linear least-squares regression, so that the estimated set of functions transforms the original data into a form in which the linear regression fits well. In addition to the principle of LKIS, an implementation using neural networks is described. Moreover, we introduce empirical performance of DMD based on the LKIS framework with several nonlinear dynamical systems and applications, which proves the feasibility of LKIS-based DMD as a fully data-driven method for modal decomposition via the Koopman operator.

## 2 Background

### 2.1 Koopman spectral analysis

We focus on a (possibly nonlinear) discrete-time autonomous dynamical system

$$\boldsymbol{x}_{t+1} = \boldsymbol{f}(\boldsymbol{x}_t), \quad \boldsymbol{x} \in \mathcal{M}, \quad t \in \mathbb{T} = \{0\} \cup \mathbb{N}, \tag{1}$$

where $\mathcal{M}$ denotes the state space and $(\mathcal{M}, \Sigma, \mu)$ represents the associated probability space. In dynamical system (1), *Koopman operator* $\mathcal{K}$ [4], [5] is defined as an infinite-dimensional linear operator that acts on *observables* $g : \mathcal{M} \to \mathbb{R}$ (or $\mathbb{C}$), i.e.,

$$\mathcal{K}g(\boldsymbol{x}) = g(\boldsymbol{f}(\boldsymbol{x})), \tag{2}$$

with which the analysis of nonlinear dynamics (1) can be lifted to a linear (but infinite-dimensional) regime. Since $\mathcal{K}$ is linear, let us consider a set of eigenfunctions $\{\varphi_1, \varphi_2, \dots\}$ of $\mathcal{K}$ with eigenvalues $\{\lambda_1, \lambda_2, \dots\}$, i.e., $\mathcal{K}\varphi_i = \lambda_i \varphi_i$ for $i \in \mathbb{N}$, where $\varphi : \mathcal{M} \to \mathbb{C}$ and $\lambda \in \mathbb{C}$. Further, suppose that $g$ can be expressed as a linear combination of those infinite number of eigenfunctions, i.e., $g(\boldsymbol{x}) = \sum_{i=1}^{\infty} \varphi_i(\boldsymbol{x})c_i$ with a set of coefficients $\{c_1, c_2, \dots\}$. By repeatedly applying $\mathcal{K}$ to both sides of this equation, we obtain the following modal decomposition:

$$g(\boldsymbol{x}_t) = \sum_{i=1}^{\infty} \lambda_i^t \varphi_i(\boldsymbol{x}_0)c_i. \tag{3}$$

Here, the value of $g$ is decomposed into a sum of *Koopman modes* $w_i = \varphi_i(\boldsymbol{x}_0)c_i$, each of which evolves over time with its frequency and decay rate respectively given by $\angle\lambda_i$ and $|\lambda_i|$, since $\lambda_i$ is a complex value. The Koopman modes and their eigenvalues can be investigated to understand the dominant characteristics of complex phenomena that follow nonlinear dynamics. The above discussion can also be applied straightforwardly to continuous-time dynamical systems [4], [5].

Modal decomposition based on $\mathcal{K}$, often referred to as Koopman spectral analysis, has been receiving attention in nonlinear physics and applied mathematics. In addition, it is a useful tool for engineering tasks including machine learning and pattern recognition; the spectra (eigenvalues) of $\mathcal{K}$ can be used as features of dynamical systems, the eigenfunctions are a useful representation of time-series for various tasks, such as regression and visualization, and $\mathcal{K}$ itself can be used for prediction and optimal control. Several methods have been proposed to compute modal decomposition based on $\mathcal{K}$, such as generalized Laplace analysis [5], [16], the Ulam–Galerkin method [17], and DMD [6], [7], [13]. DMD, which is reviewed in more detail in the next subsection, has received significant attention and been utilized in various data analysis scenarios (e.g., [6]–[9]).

Note that the Koopman operator and modal decomposition based on it can be extended to random dynamical systems actuated by process noise [4], [14], [18]. In addition, Proctor *et al.* [19], [20] discussed Koopman analysis of systems with control signals. In this paper, we primarily target autonomous deterministic dynamics (e.g., Eq. (1)) for the sake of presentation clarity.

## 2.2 Dynamic mode decomposition and Koopman invariant subspace

Let us review DMD, an algorithm for Koopman spectral analysis (further details are in the supplementary). Consider a set of observables $\{g_1, \ldots, g_n\}$ and let $\boldsymbol{g} = [g_1 \ \cdots \ g_n]^\mathsf{T}$ be a vector-valued observable. In addition, define two matrices $\boldsymbol{Y}_0, \boldsymbol{Y}_1 \in \mathbb{R}^{n \times m}$ generated by $\boldsymbol{x}_0$, $\boldsymbol{f}$ and $\boldsymbol{g}$, i.e.,

$$\boldsymbol{Y}_0 = [\boldsymbol{g}(\boldsymbol{x}_0) \ \cdots \ \boldsymbol{g}(\boldsymbol{x}_{m-1})] \quad \text{and} \quad \boldsymbol{Y}_1 = [\boldsymbol{g}(\boldsymbol{f}(\boldsymbol{x}_0)) \ \cdots \ \boldsymbol{g}(\boldsymbol{f}(\boldsymbol{x}_{m-1}))], \qquad (4)$$

where $m + 1$ is the number of snapshots in the dataset. The core functionality of DMD algorithms is computing the eigendecomposition of matrix $\boldsymbol{A} = \boldsymbol{Y}_1 \boldsymbol{Y}_0^\dagger$ [13], [21], where $\boldsymbol{Y}_0^\dagger$ is the Moore–Penrose pseudoinverse of $\boldsymbol{Y}_0$. The eigenvectors of $\boldsymbol{A}$ are referred to as *dynamic modes*, and they coincide with the Koopman modes if the corresponding eigenfunctions of $\mathcal{K}$ are in $\mathrm{span}\{g_1, \ldots, g_n\}$ [21]. Alternatively (but nearly equivalently), the condition under which DMD works as a numerical realization of Koopman spectral analysis can be described as follows.

Rather than calculating the infinite-dimensional $\mathcal{K}$ directly, we can consider the restriction of $\mathcal{K}$ to a finite-dimensional subspace. Assume the observables are elements of $L^2(\mathcal{M}, \mu)$. The *Koopman invariant subspace* is defined as $\mathcal{G} \subset L^2(\mathcal{M}, \mu)$ s.t. $\forall g \in \mathcal{G}$, $\mathcal{K}g \in \mathcal{G}$. If $\mathcal{G}$ is spanned by a finite number of functions, then the restriction of $\mathcal{K}$ to $\mathcal{G}$, which we denote $K$, becomes a *finite-dimensional linear operator*. In the sequel, we assume the existence of such $\mathcal{G}$. If $\{g_1, \ldots, g_n\}$ spans $\mathcal{G}$, then DMD's matrix $\boldsymbol{A} = \boldsymbol{Y}_1 \boldsymbol{Y}_0^\dagger$ coincides with $\boldsymbol{K} \in \mathbb{R}^{n \times n}$ asymptotically, wherein $\boldsymbol{K}$ is the realization of $K$ with regard to the frame (or basis) $\{g_1, \ldots, g_n\}$. For modal decomposition (3), the (vector-valued) Koopman modes are given by $\boldsymbol{w}$ and the values of the eigenfunctions are obtained by $\varphi = \boldsymbol{z}^\mathsf{H} \boldsymbol{g}$, where $\boldsymbol{w}$ and $\boldsymbol{z}$ are the right- and left-eigenvectors of $\boldsymbol{K}$ normalized such that $\boldsymbol{w}_i^\mathsf{H} \boldsymbol{z}_j = \delta_{i,j}$ [14], [21], and $\boldsymbol{z}^\mathsf{H}$ denotes the conjugate transpose of $\boldsymbol{z}$.

Here, an important problem in the practice of DMD arises, i.e., we often have no access to $\boldsymbol{g}$ that spans a Koopman invariant subspace $\mathcal{G}$. In this case, for nonlinear dynamics, we must manually prepare adequate observables. Several researchers have addressed this issue; Williams *et al.* [14] leveraged a dictionary of predefined basis functions to transform original data, and Kawahara [15] defined Koopman spectral analysis in a reproducing kernel Hilbert space. Brunton *et al.* [22] proposed the use of observables selected in a data-driven manner [23] from a function dictionary. Note that, for these methods, we must select an appropriate function dictionary or kernel function according to the target dynamics. However, if we have no *a priori* knowledge about them, which is often the case, such existing methods do not have to be applied successfully to nonlinear dynamics.

# 3 Learning Koopman invariant subspaces

## 3.1 Minimizing residual sum of squares of linear least-squares regression

In this paper, we propose a method to learn a set of observables $\{g_1, \ldots, g_n\}$ that spans a Koopman invariant subspace $\mathcal{G}$, given a sequence of measurements as the dataset. In the following, we summarize desirable properties for such observables, upon which the proposed method is constructed.

**Theorem 1.** *Consider a set of square-integrable observables $\{g_1, \ldots, g_n\}$, and define a vector-valued observable $\boldsymbol{g} = [g_1 \ \cdots \ g_n]^\mathsf{T}$. In addition, define a linear operator $G$ whose matrix form is given as $\boldsymbol{G} = \left( \int_{\mathcal{M}} (\boldsymbol{g} \circ \boldsymbol{f}) \boldsymbol{g}^\mathsf{H} \mathrm{d}\mu \right) \left( \int_{\mathcal{M}} \boldsymbol{g} \boldsymbol{g}^\mathsf{H} \mathrm{d}\mu \right)^\dagger$. Then, $\forall \boldsymbol{x} \in \mathcal{M}$, $\boldsymbol{g}(\boldsymbol{f}(\boldsymbol{x})) = \boldsymbol{G}\boldsymbol{g}(\boldsymbol{x})$ if and only if $\{g_1, \ldots, g_n\}$ spans a Koopman invariant subspace.*

*Proof.* If $\forall \boldsymbol{x} \in \mathcal{M}$, $\boldsymbol{g}(\boldsymbol{f}(\boldsymbol{x})) = \boldsymbol{G}\boldsymbol{g}(\boldsymbol{x})$, then for any $\hat{g} = \sum_{i=1}^n a_i g_i \in \mathrm{span}\{g_1, \ldots, g_n\}$,

$$\mathcal{K}\hat{g} = \sum_{i=1}^n a_i g_i(\boldsymbol{f}(\boldsymbol{x})) = \sum_{j=1}^n \left( \sum_{i=1}^n a_i G_{i,j} \right) g_j(\boldsymbol{x}) \in \mathrm{span}\{g_1, \ldots, g_n\},$$

where $G_{i,j}$ denotes the $(i, j)$-element of $\boldsymbol{G}$; thus, $\mathrm{span}\{g_1, \ldots, g_n\}$ is a Koopman invariant subspace. On the other hand, if $\{g_1, \ldots, g_n\}$ spans a Koopman invariant subspace, there exists a linear operator $K$ such that $\forall \boldsymbol{x} \in \mathcal{M}$, $\boldsymbol{g}(\boldsymbol{f}(\boldsymbol{x})) = \boldsymbol{K}\boldsymbol{g}(\boldsymbol{x})$; thus, $\int_{\mathcal{M}} (\boldsymbol{g} \circ \boldsymbol{f}) \boldsymbol{g}^\mathsf{H} \mathrm{d}\mu = \int_{\mathcal{M}} \boldsymbol{K} \boldsymbol{g} \boldsymbol{g}^\mathsf{H} \mathrm{d}\mu$. Therefore, an instance of the matrix form of $K$ is obtained in the form of $\boldsymbol{G}$. □

According to Theorem 1, we should obtain $\boldsymbol{g}$ that makes $\boldsymbol{g} \circ \boldsymbol{f} - \boldsymbol{G}\boldsymbol{g}$ zero. However, such problems cannot be solved with finite data because $\boldsymbol{g}$ is a function. Thus, we give the corresponding empirical

risk minimization problem based on the assumption of ergodicity of $\boldsymbol{f}$ and the convergence property of the empirical matrix as follows.

**Assumption 1.** For dynamical system (1), the time-average and space-average of a function $g : \mathcal{M} \to \mathbb{R}$ (or $\mathbb{C}$) coincide in $m \to \infty$ for almost all $\boldsymbol{x}_0 \in \mathcal{M}$, i.e.,

$$\lim_{m \to \infty} \frac{1}{m} \sum_{j=0}^{m-1} g(\boldsymbol{x}_j) = \int_{\mathcal{M}} g(\boldsymbol{x}) \mathrm{d}\mu(\boldsymbol{x}), \quad \text{for almost all } \boldsymbol{x}_0 \in \mathcal{M}.$$

**Theorem 2.** *Define $\boldsymbol{Y}_0$ and $\boldsymbol{Y}_1$ by Eq. (4) and suppose that Assumption 1 holds. If all modes are sufficiently excited in the data (i.e., $\mathrm{rank}(\boldsymbol{Y}_0) = n$), then matrix $\boldsymbol{A} = \boldsymbol{Y}_1 \boldsymbol{Y}_0^\dagger$ almost surely converges to the matrix form of linear operator $G$ in $m \to \infty$.*

*Proof.* From Assumption 1, $\frac{1}{m} \boldsymbol{Y}_1 \boldsymbol{Y}_0^\mathsf{H}$ and $\frac{1}{m} \boldsymbol{Y}_0 \boldsymbol{Y}_0^\mathsf{H}$ respectively converge to $\int_{\mathcal{M}} (\boldsymbol{g} \circ \boldsymbol{f}) \boldsymbol{g}^\mathsf{H} \mathrm{d}\mu$ and $\int_{\mathcal{M}} \boldsymbol{g} \boldsymbol{g}^\mathsf{H} \mathrm{d}\mu$ for almost all $\boldsymbol{x}_0 \in \mathcal{M}$. In addition, since the rank of $\boldsymbol{Y}_0 \boldsymbol{Y}_0^\mathsf{H}$ is always $n$, $(\frac{1}{m} \boldsymbol{Y}_0 \boldsymbol{Y}_0^\mathsf{H})^\dagger$ converges to $(\int_{\mathcal{M}} \boldsymbol{g} \boldsymbol{g}^\mathsf{H} \mathrm{d}\mu)^\dagger$ in $m \to \infty$ [24]. Consequently, in $m \to \infty$, $\boldsymbol{A} = (\frac{1}{m} \boldsymbol{Y}_1 \boldsymbol{Y}_0^\mathsf{H})(\frac{1}{m} \boldsymbol{Y}_0 \boldsymbol{Y}_0^\mathsf{H})^\dagger$ almost surely converges to $\boldsymbol{G}$, which is the matrix form of linear operator $G$. $\square$

Since $\boldsymbol{A} = \boldsymbol{Y}_1 \boldsymbol{Y}_0^\dagger$ is the minimum-norm solution of the linear least-squares regression from the columns of $\boldsymbol{Y}_0$ to those of $\boldsymbol{Y}_1$, we constitute the learning problem to estimate a set of function that transforms the original data into a form in which the linear least-squares regression fits well. In particular, we minimize RSS, which measures the discrepancy between the data and the estimated regression model (i.e., linear least-squares in this case). We define the *RSS loss* as follows:

$$\mathcal{L}_{\mathrm{RSS}}(\boldsymbol{g}; (\boldsymbol{x}_0, \ldots, \boldsymbol{x}_m)) = \left\| \boldsymbol{Y}_1 - (\boldsymbol{Y}_1 \boldsymbol{Y}_0^\dagger) \boldsymbol{Y}_0 \right\|_{\mathrm{F}}^2, \tag{5}$$

which becomes zero when $\boldsymbol{g}$ spans a Koopman invariant subspace. If we implement a smooth parametric model on $\boldsymbol{g}$, the local minima of $\mathcal{L}_{\mathrm{RSS}}$ can be found using gradient descent. We adopt $\boldsymbol{g}$ that achieves a local minimum of $\mathcal{L}_{\mathrm{RSS}}$ as a set of observables that spans (approximately) a Koopman invariant subspace.

### 3.2 Linear delay embedder for state space reconstruction

In the previous subsection, we have presented an important part of the principle of LKIS, i.e., minimization of the RSS of linear least-squares regression. Note that, to define RSS loss (5), we need access to a sequence of the original states, i.e., $(\boldsymbol{x}_0, \ldots, \boldsymbol{x}_m) \in \mathcal{M}^{m+1}$, as a dataset. In practice, however, we cannot necessarily observe full states $\boldsymbol{x}$ due to limited memory and sensor capabilities. In this case, only transformed (and possibly degenerated) measurements are available, which we denote $\boldsymbol{y} = \boldsymbol{\psi}(\boldsymbol{x})$ with a measurement function $\boldsymbol{\psi} : \mathcal{M} \to \mathbb{R}^r$. To define RSS loss (5) given only degenerated measurements, we must reconstruct the original states $\boldsymbol{x}$ from the actual observations $\boldsymbol{y}$.

Here, we utilize delay-coordinate embedding, which has been widely used for state space reconstruction in the analysis of nonlinear dynamics. Consider a univariate time-series $(\ldots, y_{t-1}, y_t, y_{t+1}, \ldots)$, which is a sequence of degenerated measurements $y_t = \psi(\boldsymbol{x}_t)$. According to the well-known Taken's theorem [25], [26], a faithful representation of $\boldsymbol{x}_t$ that preserves the structure of the state space can be obtained by $\tilde{\boldsymbol{x}}_t = \begin{bmatrix} y_t & y_{t-\tau} & \cdots & y_{t-(d-1)\tau} \end{bmatrix}^\mathsf{T}$ with some lag parameter $\tau$ and embedding dimension $d$ if $d$ is greater than $2 \dim(\boldsymbol{x})$. For a multivariate time-series, embedding with non-uniform lags provides better reconstruction [27]. For example, when we have a two-dimensional time-series $\boldsymbol{y}_t = \begin{bmatrix} y_{1,t} & y_{2,t} \end{bmatrix}^\mathsf{T}$, an embedding with non-uniform lags is similar to $\tilde{\boldsymbol{x}}_t = \begin{bmatrix} y_{1,t} & y_{1,t-\tau_{11}} & \cdots & y_{1,t-\tau_{1d_1}} & y_{2,t} & y_{2,t-\tau_{21}} & \cdots & y_{2,t-\tau_{2d_2}} \end{bmatrix}^\mathsf{T}$ with each value of $\tau$ and $d$. Several methods have been proposed for selection of $\tau$ and $d$ [27]–[29]; however, appropriate values may depend on the given application (attractor inspection, prediction, etc.).

In this paper, we propose to surrogate the parameter selection of the delay-coordinate embedding by learning a *linear delay embedder* from data. Formally, we learn embedder $\phi$ such that

$$\tilde{\boldsymbol{x}}_t = \phi(\boldsymbol{y}_t^{(k)}) = \boldsymbol{W}_\phi \begin{bmatrix} \boldsymbol{y}_t^\mathsf{T} & \boldsymbol{y}_{t-1}^\mathsf{T} & \cdots & \boldsymbol{y}_{t-k+1}^\mathsf{T} \end{bmatrix}^\mathsf{T}, \quad \boldsymbol{W}_\phi \in \mathbb{R}^{p \times kr}, \tag{6}$$

where $p = \dim(\tilde{\boldsymbol{x}})$, $r = \dim(\boldsymbol{y})$, and $k$ is a hyperparameter of maximum lag. We estimate weight $\boldsymbol{W}_\phi$ as well as the parameters of $\boldsymbol{g}$ by minimizing RSS loss (5), which is now defined using $\tilde{\boldsymbol{x}}$ instead of $\boldsymbol{x}$. Learning $\phi$ from data yields an embedding that is suitable for learning a Koopman invariant subspace. Moreover, we can impose L1 regularization on weight $\boldsymbol{W}_\phi$ to make it highly interpretable if necessary according to the given application.

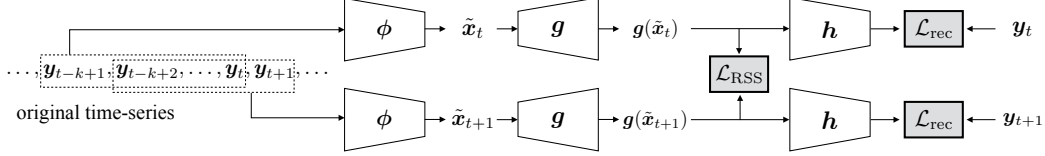

Figure 1: An instance of LKIS framework, in which $\boldsymbol{g}$ and $\boldsymbol{h}$ are implemented by MLPs.

### 3.3 Reconstruction of original measurements

Simple minimization of $\mathcal{L}_{\text{RSS}}$ may yield trivial $\boldsymbol{g}$, such as constant values. We should impose some constraints to prevent such trivial solutions. In the proposed framework, modal decomposition is first obtained in terms of learned observables $\boldsymbol{g}$; thus, the values of $\boldsymbol{g}$ must be back-projected to the space of the original measurements $\boldsymbol{y}$ to obtain a physically meaningful representation of the dynamic modes. Therefore, we modify the loss function by employing an additional term such that the original measurements $\boldsymbol{y}$ can be reconstructed from the values of $\boldsymbol{g}$ by a *reconstructor* $\boldsymbol{h}$, i.e., $\boldsymbol{y} \approx \boldsymbol{h}(\boldsymbol{g}(\tilde{\boldsymbol{x}}))$. Such term is given as follows:

$$\mathcal{L}_{\text{rec}}(\boldsymbol{h}, \boldsymbol{g}; (\tilde{\boldsymbol{x}}_0, \ldots, \tilde{\boldsymbol{x}}_m)) = \sum_{j=0}^{m} \|\boldsymbol{y}_j - \boldsymbol{h}(\boldsymbol{g}(\tilde{\boldsymbol{x}}_j))\|^2, \tag{7}$$

and, if $\boldsymbol{h}$ is a smooth parametric model, this term can also be reduced using gradient descent. Finally, the objective function to be minimized becomes

$$\mathcal{L}(\boldsymbol{\phi}, \boldsymbol{g}, \boldsymbol{h}; (\boldsymbol{y}_0, \ldots, \boldsymbol{y}_m)) = \mathcal{L}_{\text{RSS}}(\boldsymbol{g}, \boldsymbol{\phi}; (\tilde{\boldsymbol{x}}_{k-1}, \ldots, \tilde{\boldsymbol{x}}_m)) + \alpha \mathcal{L}_{\text{rec}}(\boldsymbol{h}, \boldsymbol{g}; (\tilde{\boldsymbol{x}}_{k-1}, \ldots, \tilde{\boldsymbol{x}}_m)), \tag{8}$$

where $\alpha$ is a parameter that controls the balance between $\mathcal{L}_{\text{RSS}}$ and $\mathcal{L}_{\text{rec}}$.

### 3.4 Implementation using neural networks

In Sections 3.1–3.3, we introduced the main concepts for the LKIS framework, i.e., RSS loss minimization, learning the linear delay embedder, and reconstruction of the original measurements. Here, we demonstrate an implementation of the LKIS framework using neural networks.

Figure 1 shows a schematic diagram of the implementation of the framework. We model $\boldsymbol{g}$ and $\boldsymbol{h}$ using multi-layer perceptrons (MLPs) with a parametric ReLU activation function [30]. Here, the sizes of the hidden layer of MLPs are defined by the arithmetic means of the sizes of the input and output layers of the MLPs. Thus, the remaining tunable hyperparameters are $k$ (maximum delay of $\boldsymbol{\phi}$), $p$ (dimensionality of $\tilde{\boldsymbol{x}}$), and $n$ (dimensionality of $\boldsymbol{g}$). To obtain $\boldsymbol{g}$ with dimensionality much greater than that of the original measurements, we found that it was useful to set $k > 1$ even when full-state measurements (e.g., $\boldsymbol{y} = \boldsymbol{x}$) were available.

After estimating the parameters of $\boldsymbol{\phi}$, $\boldsymbol{g}$, and $\boldsymbol{h}$, DMD can be performed normally by using the values of the learned $\boldsymbol{g}$, defining the data matrices in Eq. (4), and computing the eigendecomposition of $\boldsymbol{A} = \boldsymbol{Y}_1 \boldsymbol{Y}_0^\dagger$; the dynamic modes are obtained by $\boldsymbol{w}$, and the values of the eigenfunctions are obtained by $\varphi = \boldsymbol{z}^{\mathsf{H}} \boldsymbol{g}$, where $\boldsymbol{w}$ and $\boldsymbol{z}$ are the right- and left-eigenvectors of $\boldsymbol{A}$. See Section 2.2 for details.

In the numerical experiments described in Sections 5 and 6, we performed optimization using first-order gradient descent. To stabilize optimization, batch normalization [31] was imposed on the inputs of hidden layers. Note that, since RSS loss function (5) is *not* decomposable with regard to data points, convergence of stochastic gradient descent (SGD) cannot be shown straightforwardly. However, we empirically found that the non-decomposable RSS loss was often reduced successfully, even with mini-batch SGD. Let us show an example; the full-batch RSS loss (denoted $\mathcal{L}_{\text{RSS}}^\star$) under the updates of the mini-batch SGD are plotted in the rightmost panel of Figure 4. Here, $\mathcal{L}_{\text{RSS}}^\star$ decreases rapidly and remains small. For SGD on non-decomposable losses, Kar *et al.* [32] provided guarantees for some cases; however, examining the behavior of more general non-decomposable losses under mini-batch updates remains an open problem.

## 4 Related work

The proposed framework is motivated by the operator-theoretic view of nonlinear dynamical systems. In contrast, learning a generative (state-space) model for nonlinear dynamical systems directly has been actively studied in machine learning and optimal control communities, on which we mention a

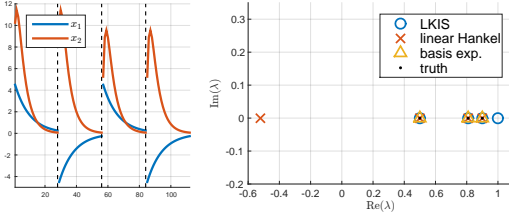
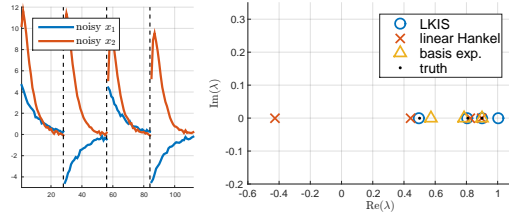

Figure 2: (*left*) Data generated from system (9) and (*right*) the estimated Koopman eigenvalues. While linear Hankel DMD produces an inconsistent eigenvalue, LKIS-DMD successfully identifies $\lambda$, $\mu$, $\lambda^2$, and $\lambda^0\mu^0 = 1$.

Figure 3: (*left*) Data generated from system (9) and white Gaussian observation noise and (*right*) the estimated Koopman eigenvalues. LKIS-DMD successfully identifies the eigenvalues even with the observation noise.

few examples. A classical but popular method for learning nonlinear dynamical systems is using an expectation-maximization algorithm with Bayesian filtering/smoothing (see, e.g., [33]). Recently, using approximate Bayesian inference with the variational autoencoder (VAE) technique [34] to learn generative dynamical models has been actively researched. Chung *et al.* [35] proposed a recurrent neural network with random latent variables, Gao *et al.* [36] utilized VAE-based inference for neural population models, and Johnson *et al.* [37] and Krishnan *et al.* [38] developed inference methods for structured models based on inference with a VAE. In addition, Karl *et al.* [39] proposed a method to obtain a more consistent estimation of nonlinear state space models. Moreover, Watter *et al.* [40] proposed a similar approach in the context of optimal control. Since generative models are intrinsically aware of process and observation noises, incorporating methodologies developed in such studies to the operator-theoretic perspective is an important open challenge to explicitly deal with uncertainty.

We would like to mention some studies closely related to our method. After the first submission of this manuscript (in May 2017), several similar approaches to learning data transform for Koopman analysis have been proposed [41]–[45]. The relationships and relative advantages of these methods should be elaborated in the future.

## 5 Numerical examples

In this section, we provide numerical examples of DMD based on the LKIS framework (LKIS-DMD) implemented using neural networks. We conducted experiments on three typical nonlinear dynamical systems: a fixed-point attractor, a limit-cycle attractor, and a system with multiple basins of attraction. We show the results of comparisons with other recent DMD algorithms, i.e., Hankel DMD [46], [47], extended DMD [14], and DMD with reproducing kernels [15]. The detailed setups of the experiments discussed in this section and the next section are described in the supplementary.

**Fixed-point attractor**   Consider a two-dimensional nonlinear map on $\boldsymbol{x}_t = [x_{1,t} \quad x_{2,t}]^\mathsf{T}$:
$$x_{1,t+1} = \lambda x_{1,t}, \quad x_{2,t+1} = \mu x_{2,t} + (\lambda^2 - \mu)x_{1,t}^2, \tag{9}$$
which has a stable equilibrium at the origin if $\lambda, \mu < 1$. The Koopman eigenvalues of system (9) include $\lambda$ and $\mu$, and the corresponding eigenfunctions are $\varphi_\lambda(\boldsymbol{x}) = x_1$ and $\varphi_\mu(\boldsymbol{x}) = x_2 - x_1^2$, respectively. $\lambda^i\mu^j$ is also an eigenvalue with corresponding eigenfunction $\varphi_\lambda^i\varphi_\mu^j$. A minimal Koopman invariant subspace of system (9) is $\mathrm{span}\{x_1, x_2, x_1^2\}$, and the eigenvalues of the Koopman operator restricted to such subspace include $\lambda$, $\mu$ and $\lambda^2$. We generated a dataset using system (9) with $\lambda = 0.9$ and $\mu = 0.5$ and applied LKIS-DMD ($n = 4$), linear Hankel DMD [46], [47] (delay 2), and DMD with basis expansion by $\{x_1, x_2, x_1^2\}$, which corresponds to extended DMD [14] with a right and minimal observable dictionary. The estimated Koopman eigenvalues are shown in Figure 2, wherein LKIS-DMD successfully identifies the eigenvalues of the target invariant subspace. In Figure 3, we show eigenvalues estimated using data contaminated with white Gaussian observation noise ($\sigma = 0.1$). The eigenvalues estimated by LKIS-DMD coincide with the true values even with the observation noise, whereas the results of DMD with basis expansion (i.e., extended DMD) are directly affected by the observation noise.

**Limit-cycle attractor**   We generated data from the limit cycle of the FitzHugh–Nagumo equation
$$\dot{x_1} = x_1^3/3 + x_1 - x_2 + I, \quad \dot{x_2} = c(x_1 - bx_2 + a), \tag{10}$$
where $a = 0.7$, $b = 0.8$, $c = 0.08$, and $I = 0.8$. Since trajectories in a limit-cycle are periodic, the (discrete-time) Koopman eigenvalues should lie near the unit circle. Figure 4 shows the eigenvalues

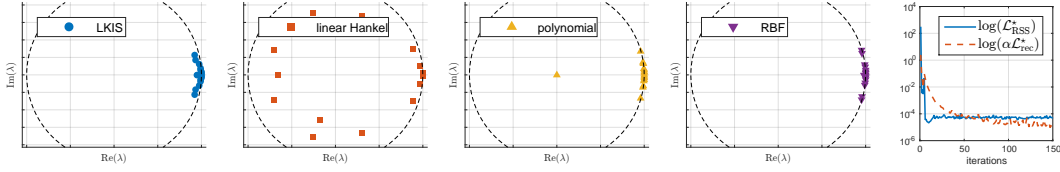

Figure 4: The left four panels show the estimated Koopman eigenvalues on the limit-cycle of the FitzHugh-Nagumo equation by LKIS-DMD, linear Hankel DMD, and kernel DMDs with polynomial and RBF kernels. The hyperparameters of each DMD are set to produce 16 eigenvalues. The rightmost plot shows the full-batch (size 2,000) loss under mini-batch (size 200) SGD updates along iterations. Non-decomposable part $\mathcal{L}_{\mathrm{RSS}}^{\star}$ decreases rapidly and remains small, even by SGD.

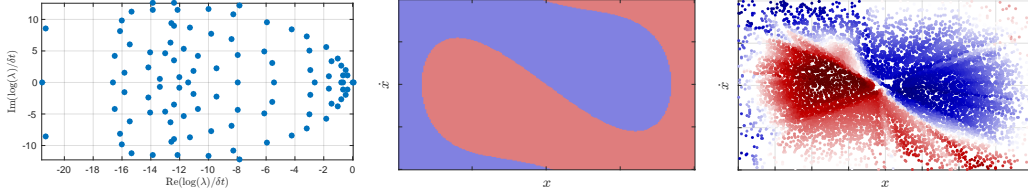

Figure 5: (*left*) The continuous-time Koopman eigenvalues estimated by LKIS-DMD on the Duffing equation. (*center*) The true basins of attraction of the Duffing equation, wherein points in the blue region evolve toward $(1, 0)$ and points in the red region evolve toward $(-1, 0)$. Note that the stable manifold of the saddle point is not drawn precisely. (*right*) The values of the Koopman eigenfunction with a nearly zero eigenvalue computed by LKIS-DMD, whose level sets should correspond to the basins of attraction. There is rough agreement between the true boundary of the basins of attraction and the numerically computed boundary. The right two plots are best viewed in color.

estimated by LKIS-DMD ($n = 16$), linear Hankel DMD [46], [47] (delay 8), and DMDs with reproducing kernels [15] (polynomial kernel of degree 4 and RBF kernel of width 1). The eigenvalues produced by LKIS-DMD agree well with those produced by kernel DMDs, whereas linear Hankel DMD produces eigenvalues that would correspond to rapidly decaying modes.

**Multiple basins of attraction**     Consider the unforced Duffing equation

$$\ddot{x} = -\delta\dot{x} - x(\beta + \alpha x^2), \quad \boldsymbol{x} = [x \quad \dot{x}]^{\mathsf{T}}, \tag{11}$$

where $\alpha = 1$, $\beta = -1$, and $\delta = 0.5$. States $\boldsymbol{x}$ following (11) evolve toward $[1 \quad 0]^{\mathsf{T}}$ or $[-1 \quad 0]^{\mathsf{T}}$ depending on which basin of attraction the initial value belongs to unless the initial state is on the stable manifold of the saddle. Generally, a Koopman eigenfunction whose continuous-time eigenvalue is zero takes a constant value in each basin of attraction [14]; thus, the contour plot of such an eigenfunction shows the boundary of the basins of attraction. We generated 1,000 episodes of time-series starting at different initial values uniformly sampled from $[-2, 2]^2$. The left plot in Figure 5 shows the continuous-time Koopman eigenvalues estimated by LKIS-DMD ($n = 100$), all of which correspond to decaying modes (i.e., negative real parts) and agree with the property of the data. The center plot in Figure 5 shows the true basins of attraction of (11), and the right plot shows the estimated values of the eigenfunction corresponding to the eigenvalue of the smallest magnitude. The surface of the estimated eigenfunction agrees qualitatively with the true boundary of the basins of attractions, which indicates that LKIS-DMD successfully identifies the Koopman eigenfunction.

## 6   Applications

The numerical experiments in the previous section demonstrated the feasibility of the proposed method as a fully data-driven method for Koopman spectral analysis. Here, we introduce practical applications of LKIS-DMD.

**Chaotic time-series prediction**     Prediction of a chaotic time-series has received significant interest in nonlinear physics. We would like to perform the prediction of a chaotic time-series using DMD, since DMD can be naturally utilized for prediction as follows. Since $\boldsymbol{g}(\boldsymbol{x}_t)$ is decomposed as $\sum_{i=1}^{n} \varphi_i(\boldsymbol{x}_t)\boldsymbol{c}_i$ and $\varphi$ is obtained by $\varphi_i(\boldsymbol{x}_t) = \boldsymbol{z}_i^{\mathsf{H}}\boldsymbol{g}(\boldsymbol{x}_t)$ where $\boldsymbol{z}_i$ is a left-eigenvalue of $\boldsymbol{K}$, the next step of $\boldsymbol{g}$ can be described in terms of the current step, i.e., $\boldsymbol{g}(\boldsymbol{x}_{t+1}) = \sum_{i=1}^{n} \lambda_i(\boldsymbol{z}_i^{\mathsf{H}}\boldsymbol{g}(\boldsymbol{x}_t))\boldsymbol{c}_i$. In

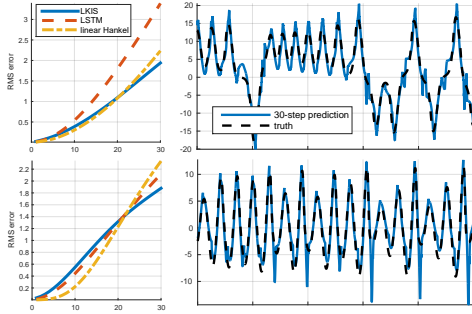

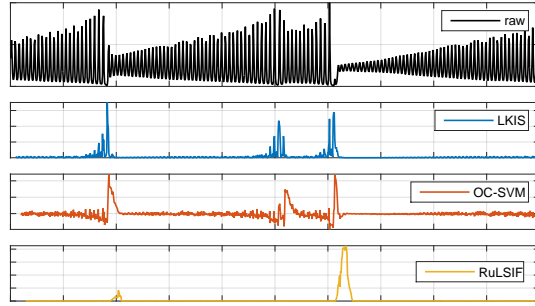

Figure 6: The left plot shows RMS errors from 1- to 30-step predictions, and the right plot shows a part of the 30-step prediction obtained by LKIS-DMD on (*upper*) the Lorenz-$x$ series and (*lower*) the Rossler-$x$ series.

Figure 7: The top plot shows the raw time-series obtained by a far-infrared laser [50]. The other plots show the results of unstable phenomena detection, wherein the peaks should correspond to the occurrences of unstable phenomena.

addition, in the case of LKIS-DMD, the values of $g$ must be back-projected to $y$ using the learned $h$. We generated two types of univariate time-series by extracting the $\{x\}$ series of the Lorenz attractor [48] and the Rossler attractor [49]. We simulated 25,000 steps for each attractor and used the first 10,000 steps for training, the next 5,000 steps for validation, and the last 10,000 steps for testing prediction accuracy. We examined the prediction accuracy of LKIS-DMD, a simple LSTM network, and linear Hankel DMD [46], [47], all of whose hyperparameters were tuned using the validation set. The prediction accuracy of every method and an example of the predicted series on the test set by LKIS-DMD are shown in Figure 6. As can be seen, the proposed LKIS-DMD achieves the smallest root-mean-square (RMS) errors in the 30-step prediction.

**Unstable phenomena detection**    One of the most popular applications of DMD is the investigation of the global characteristics of dynamics by inspecting the spatial distribution of the dynamic modes. In addition to the spatial distribution, we can investigate the temporal profiles of mode activations by examining the values of corresponding eigenfunctions. For example, assume there is an eigenfunction $\varphi_{\lambda \ll 1}$ that corresponds to a discrete-time eigenvalue $\lambda$ whose magnitude is considerably smaller than one. Such a small eigenvalue indicates a rapidly decaying (i.e., unstable) mode; thus, we can detect occurrences of unstable phenomena by observing the values of $\varphi_{\lambda \ll 1}$. We applied LKIS-DMD ($n = 10$) to a time-series generated by a far-infrared laser, which was obtained from the Santa Fe Time Series Competition Data [50]. We investigated the values of eigenfunction $\varphi_{\lambda \ll 1}$ corresponding to the eigenvalue of the smallest magnitude. The original time-series and values of $\varphi_{\lambda \ll 1}$ obtained by LKIS-DMD are shown in Figure 7. As can be seen, the activations of $\varphi_{\lambda \ll 1}$ coincide with sudden decays of the pulsation amplitudes. For comparison, we applied the novelty/change-point detection technique using one-class support vector machine (OC-SVM) [51] and direct density-ratio estimation by relative unconstrained least-squares importance fitting (RuLSIF) [52]. We computed AUC, defining the sudden decays of the amplitudes as the points to be detected, which were 0.924, 0.799, and 0.803 for LKIS, OC-SVM, and RuLSIF, respectively.

# 7   Conclusion

In this paper, we have proposed a framework for learning Koopman invariant subspaces, which is a fully data-driven numerical algorithm for Koopman spectral analysis. In contrast to existing approaches, the proposed method learns (approximately) a Koopman invariant subspace entirely from the available data based on the minimization of RSS loss. We have shown empirical results for several typical nonlinear dynamics and application examples.

We have also introduced an implementation using multi-layer perceptrons; however, one possible drawback of such an implementation is the local optima of the objective function, which makes it difficult to assess the adequacy of the obtained results. Rather than using neural networks, the observables to be learned could be modeled by a sparse combination of basis functions as in [23] but still utilizing optimization based on RSS loss. Another possible future research direction could be incorporating approximate Bayesian inference methods, such as VAE [34]. The proposed framework is based on a discriminative viewpoint, but inference methodologies for generative models could be used to modify the proposed framework to explicitly consider uncertainty in data.

**Acknowledgments**

This work was supported by JSPS KAKENHI Grant No. JP15J09172, JP26280086, JP16H01548, and JP26289320.

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
