[Supplementary Material]

# Supplementary Material for "Learning Koopman Invariant Subspaces for Dynamic Mode Decomposition," NIPS 2017

## A   Algorithm of dynamic mode decomposition

Dynamic mode decomposition (DMD) was originally invented as a tool for inspecting fluid flows [1], [2], and it has been utilized in various fields other than fluid dynamics. An output of DMD coincides with Koopman spectral analysis if we have $\boldsymbol{g}$ that spans a Koopman invariant subspace. The popular algorithm of DMD [3], which is based on the singular value decomposition (SVD) of a data matrix, is defined as follows.

**Algorithm** (DMD [3])**.**

(1) Given a sequence of $\boldsymbol{g}(\boldsymbol{x})$, define data matrices $\boldsymbol{Y}_0 = [\boldsymbol{g}(\boldsymbol{x}_0) \quad \cdots \quad \boldsymbol{g}(\boldsymbol{x}_{m-1})]$ and $\boldsymbol{Y}_1 = [\boldsymbol{g}(\boldsymbol{f}(\boldsymbol{x}_0)) \quad \cdots \quad \boldsymbol{g}(\boldsymbol{f}(\boldsymbol{x}_{m-1}))]$.
(2) Calculate the compact SVD of $\boldsymbol{Y}_0$ as $\boldsymbol{Y}_0 = \boldsymbol{U}_r \boldsymbol{S}_r \boldsymbol{V}_r^{\mathsf{H}}$, where $r$ is the rank of $\boldsymbol{Y}_0$.
(3) Compute a matrix $\tilde{\boldsymbol{A}} = \boldsymbol{U}_r^{\mathsf{H}} \boldsymbol{Y}_1 \boldsymbol{V}_r \boldsymbol{S}_r^{-1}$.
(4) Calculate eigendecomposition of $\tilde{\boldsymbol{A}}$, i.e., compute $\tilde{\boldsymbol{w}}$ and $\lambda$ such that $\tilde{\boldsymbol{A}}\tilde{\boldsymbol{w}} = \lambda\tilde{\boldsymbol{w}}$.
(5) In addition, calculate left-eigenvectors $\tilde{\boldsymbol{z}}$ of $\tilde{\boldsymbol{A}}$.
(6) Back-project the eigenvectors to the original space by $\boldsymbol{w} = \lambda^{-1}\boldsymbol{Y}_1 \boldsymbol{V}_r \boldsymbol{S}_r^{-1}\tilde{\boldsymbol{w}}$ and $\boldsymbol{z} = \boldsymbol{U}_r\tilde{\boldsymbol{z}}$.
(7) Normalize $\boldsymbol{w}$ and $\boldsymbol{z}$ such that $\boldsymbol{w}_i^{\mathsf{H}}\boldsymbol{z}_j = \delta_{i,j}$, where $\delta_{i,j} = 1$ if $i = j$ and $\delta_{i,j} = 0$ otherwise.
(8) Return *dynamic modes* $\boldsymbol{w}$ and corresponding eigenvalues $\lambda$. In addition, return the values of corresponding eigenfunctions by $\varphi = \boldsymbol{z}^{\mathsf{H}}\boldsymbol{g}$.

The eigenvalues computed using the algorithm above are "discrete-time" ones, in the sense that they represent frequencies and decay rates in terms of discrete-time dynamical systems. The "continuous-time" counterparts can be computed easily by $\lambda_c = \log(\lambda)/\Delta t$, where $\Delta t$ is the time interval in the discrete-time setting.

Note that the definition above presumes the access to an appropriate observable $\boldsymbol{g}$. In contrast, in the proposed method, the above-mentioned algorithm is run *after* applying the proposed framework to learn $\boldsymbol{g}$ from observed data.

## B   Detailed experimental setup

In this supplementary section, the configurations of the numerical examples and applications, which were omitted in the main text, are described.

### B.1   General settings

**Hyperparameters**    In each experiment, parameter $\alpha$ was fixed at $0.01$. Note that the quality of the results was not sensitive to the values of $\alpha$. We modeled $\boldsymbol{g}$ and $\boldsymbol{h}$ with multi-layer perceptrons by setting the number of hidden nodes (denoted $n_h$) as the arithmetic means of the input and output sizes, i.e., $n_h = \mathrm{round}((p + n)/2)$ for $\boldsymbol{g}$ and $n_h = \mathrm{round}((n + r)/2)$ for $\boldsymbol{h}$, where $r = \dim(\boldsymbol{y})$, $p = \dim(\tilde{\boldsymbol{x}})$, and $n = \dim(\boldsymbol{g})$. Therefore, the remaining hyperparameters to be tuned were $k$

(maximum lag), $p$, and $n$. However, unless otherwise noted, we fixed $p$ by $p = kr$. Consequently, the independent hyperparameters were $k$ and $n$.

**Preprocessing** One *must not subtract the mean* from the original data because subtracting something from the data may change the spectra of the underlying dynamical systems (see, e.g., [4]). If the absolute values of the data were too large, we simply divided the data by the maximum absolute value for each series.

**Optimization** In optimization, we found that the adaptive learning rate by SMORMS3 [5] achieved fast convergence compared to a fixed learning rate and other adaptation techniques. The maximum learning rate of SMORMS3 was selected from $10^{-3}$ to $10^{-2}$ in each experiment according to the amount of data. In some cases, optimization was performed in two stages: the parameters of $\phi$, $\boldsymbol{g}$, and $\boldsymbol{h}$ were updated in the first stage, and, in the second stage, the parameters of $\phi$ and $\boldsymbol{g}$ were fixed and only $\boldsymbol{h}$ was updated. This two-stage optimization was particularly useful for the application of prediction, where a precise reconstruction of the original measurements was necessary. Moreover, when the original states $\boldsymbol{x}$ of the dynamical system were available and used without delay (i.e., $k = 1$ and $p = r$), parameter $\boldsymbol{W}_\phi$ of the linear embedder was fixed to be an identity matrix (i.e., no embedder was used). Also, we set the mini-batch size from 100 to 500 because smaller mini-batches often led to an unstable computation of pseudo-inverse.

## B.2 Fixed-point attractor experiment

In the experiment using the fixed-point attractor, the data were generated with four initial values: $\begin{bmatrix} 5 & 5 \end{bmatrix}^\mathsf{T}$, $\begin{bmatrix} -5 & 5 \end{bmatrix}^\mathsf{T}$, $\begin{bmatrix} 5 & -5 \end{bmatrix}^\mathsf{T}$, and $\begin{bmatrix} -5 & 5 \end{bmatrix}^\mathsf{T}$, with the length of each episode being 30. In the case of noisy dataset, the standard deviation of the observation noise was set to 0.1. In both experiments (with and without observation noise), we set $k = 2$ and $n = 4$ to cover the minimal three-dimensional Koopman invariant subspace.

## B.3 Limit-cycle attractor experiment

The data were generated using MATLAB's `ode45` function [6], which was run with time-step $\Delta t = 0.1$ and initial value $\boldsymbol{x}_0 = \begin{bmatrix} 1 & 1.6 \end{bmatrix}^\mathsf{T}$ for 2,000 steps. The hyperparameters of LKIS-DMD, linear Hankel DMD, and kernel DMDs were set such that they produced 16 eigenvalues, i.e., $k = 8$ and $n = 16$ for LKIS-DMD, and POD modes whose singular value was less than $\varepsilon$ were disposed in kernel DMDs ($\varepsilon = 0.0001$ for the polynomial kernel and $\varepsilon = 0.05$ for the RBF kernel).

## B.4 Multiple basins of attraction experiment

The data were generated using the settings provided in the literature [7]; 1,000 initial values were drawn from the uniform distribution on $[-2, 2] \times [-2, 2]$ and each initial value was proceeded in time for 11 steps with $\Delta t = 0.25$. We used MATLAB's `ode45` function for numerical integration. For LKIS-DMD, we set $k = 1$ and $n = 100$. Note that the values of the estimated eigenfunction were evaluated and plotted in consideration of each data point.

## B.5 Chaotic time-series prediction experiment

The data were generated from the Lorenz attractor [8] (parameters $\beta = 8/3$, $\sigma = 10$, and $\rho = 28$) and the Rossler attractor [9] (parameters $a = 0.2$, $b = 0.2$, and $c = 5.7$). We generated 25,000 steps for each attractor and divided them into training, validation, and test sets. For all methods, the delay dimension was fixed at 7, i.e., $k = 7$ for LKIS-DMD and linear Hankel DMD, and backpropagation was truncated to length 7 to learn the LSTM network. We tuned $n$ of LKIS-DMD and the dimensionality of LSTM's hidden state (denoted $n_h$) according to the 30-step prediction accuracies obtained using the validation set. Here, we obtained $n = 5$ and $n_h = 5$ for the Lorenz data and $n = 6$ and $n_h = 3$ for the Rossler data.

In this experiment, LSTM was applied because it had been utilized for various nonlinear time-series, and Hankel DMD was used because it had been successfully utilized for analysis of chaotic systems [10].

## B.6   Unstable phenomena detection experiment

The dataset was obtained from the Santa Fe Time Series Competition Data [11]. Note that the author's [11] original web page was not available on the date of submission of this manuscript (May 2017); however, the dataset itself was still available online. The length of delay (or sliding window) was fixed to 10 for all methods applied in this experiment. In addition, no intensive tuning of the other hyperparameters was conduct because the purpose was qualitative. The default settings of `libsvm` [12] were used for the one-class SVM (except for $\nu = 0.05$). For the density-ratio estimation by RuLSIF, the default values of the implementation by the authors of [13] were used.

In this experiment, OC-SVM was applied because it was a kind of *de facto* standard for novelty/change-point detection, and RuLSIF ws used because it had achieved the best performance among methods based on density-ratio estimation [13].