[Reviews · NeurIPS 2017]

Reviewer 1



The authors proposed a fully data-driven method for Koopman analysis learning invariant subspaces. The authors proposed a standard linear regression with additional constrains that are implemented using a Neural Network to estimate the Koopman modes. The authors nicely review the main concepts of Koopman operator and dynamic mode decomposition. They proposed two costs functions; the first one finding the parameter $g$ that minimizes the error between data and estimated regression model and another one minimizing the observations $y$. There is however some issues that need further clarifications. Eq (6) is used to find the embedding between $\bold y$ and $\tilde x$ and this is included in eq (5). What is the actual form of the minimization? This is just verbally stated but is not clear how the minimization has been modified. One major point that needs to be clarified is the coupling of network itself because g and h are coupled. It is however not clear if this are decomposed as a single network or 2 networks. In the later case, is this trained end-to-end?. What are the backpropagation rules for Eq (7)? Another minor comments is with regards to Taken's theorem, the authors must state what do they mean by "faithful representation." since that term do not have any scientific meaning.

Reviewer 2



This paper presents a method that takes advantage of MLPs to learning non-linear functions that support Koopman analysis of time-series data. The framework is based on Koopman operator theory, an observation that complex, nonlinear dynamics may be embedded with nonlinear functions g where a spectral properties of a linear operator K can be informative about both the dynamics of the system and predict future data. The paper proposes a neural network architecture and a set of loss functions suitable for learning this embedding in a Koopman invariant subspace, directly from data. The method is demonstrate both on numerical examples and on a few applications (Lorenz, Rossler, and unstable phenomena data). The paper is exceptionally clearly written, and the use of a neural network for finding Koopman invariant subspaces, a challenging and widely applicable task, is well motivated. The figures nicely illustrate the results, and I find the examples convincing. There's a few questions I was curious about. Firstly, how well does LKIS tolerate larger magnitudes of noise? Does it converge on the right Koopman eigenfunctions, or is there a point past which it fails? Second, how do the g's found by LKIS compare with those found by EDMD and kernel DMD? I read through the author’s rebuttal and my colleagues’ review, and decided to not change my assessment of the paper.

Reviewer 3



This paper proposed a data-driven method for learning nonlinear systems (both observation functions and their eigen/spectral representations) using the tool of Koopman spectral analysis. The authors proposed to minimize the RSS square loss within a LS-regression framework. In addition, the author also proposed to use MLP/ANN as the parametric function family that transform the data into a linear space. The proposed algorithm was evaluated on several synthetic non-linear models and real world unstable time series. * The selection of compared algorithms in section 5/6 seems rather basic or random. It is not quite clear why such baselines are competitive or state-of-the-art. * For the Santa Fe dataset, since OC-SVM is used, would it be possible to provide some metric other than the qualitative estimation? For example, AUC, Precision, etc. * (correct me if I am wrong) I could not locate how the hyper/free parameters (k, p, n) should be tuned in practice. It is not clear what are the optimal choice of these free parameters in the provided examples. Meanwhile, more details on the SGD parameters are not clear.